# Nicotinamide Mononucleotide (NMN) Ameliorates Free Fatty Acid-Induced Pancreatic β-Cell Dysfunction via the NAD^+^/AMPK/SIRT1/HIF-1α Pathway

**DOI:** 10.3390/ijms251910534

**Published:** 2024-09-30

**Authors:** Yan Wang, Si Liu, Linyao Ying, Keyi Zhang, Hao Li, Na Liang, Lin Xiao, Gang Luo

**Affiliations:** Xiangya School of Public Health, Central South University, Changsha 410078, China; estherwyan@csu.edu.cn (Y.W.); 226911029@csu.edu.cn (S.L.); 226912059@csu.edu.cn (L.Y.); zhangky@csu.edu.cn (K.Z.); 236912060@csu.edu.cn (H.L.); 236912047@csu.edu.cn (N.L.); xiaolinxl@csu.edu.cn (L.X.)

**Keywords:** pancreatic β-cell dysfunction, free fatty acid, nicotinamide mononucleotide, NAD^+^, HIF-1α

## Abstract

As the sole producers of insulin under physiological conditions, the normal functioning of pancreatic β cells is crucial for maintaining glucose homeostasis in the body. Due to the high oxygen and energy demands required for insulin secretion, hypoxia has been shown to play a critical role in pancreatic β-cell dysfunction. Lipid metabolism abnormalities, a common metabolic feature in type 2 diabetic patients, are often accompanied by tissue hypoxia caused by metabolic overload and lead to increased free fatty acid (FFA) levels. However, the specific mechanisms underlying FFA-induced β-cell dysfunction remain unclear. Nicotinamide mononucleotide (NMN), a naturally occurring bioactive nucleotide, has garnered significant attention in recent years for its effectiveness in replenishing NAD^+^ and alleviating various diseases. Nevertheless, studies exploring the mechanisms through which NMN influences β-cell dysfunction remain scarce. In this study, we established an in vitro β-cell dysfunction model by treating INS-1 cells with palmitate (PA), including control, PA-treated, and PA combined with NMN or activator/inhibitor groups. Compared to the control group, cells treated with PA alone showed significantly reduced insulin secretion capacity and decreased expression of proteins related to the NAD^+^/AMPK/SIRT1/HIF-1α pathway. In contrast, NMN supplementation significantly restored the expression of pathway-related proteins by activating NAD^+^ and effectively improved insulin secretion. Results obtained using HIF-1α and AMPK inhibitors/activators further supported these findings. In conclusion, our study demonstrates that NMN reversed the PA-induced downregulation of the NAD^+^/AMPK/SIRT1/HIF-1α pathway, thereby alleviating β-cell dysfunction. Our study investigated the mechanisms underlying PA-induced β-cell dysfunction, examined how NMN mitigates this dysfunction and offered new insights into the therapeutic potential of NMN for treating β-cell dysfunction and T2DM.

## 1. Introduction

Type 2 diabetes mellitus (T2DM) is a persistent metabolic condition marked by elevated blood glucose levels. Pancreatic β cells, being the only producers of insulin under normal conditions, are essential for maintaining glucose homeostasis in the body [1]. In the initial stages of T2DM, β-cell proliferation and improved function can compensate for insulin resistance, helping to maintain normal blood glucose levels. However, as the disease progresses and metabolic overload persists, β-cell dysfunction occurs [2]. Pioneering studies indicate that at the onset of T2DM, β-cell function is already diminished to 50–80% of normal capacity [3,4]. Regardless of the specific therapeutic interventions employed, impaired β-cell function at this stage is linked to poor glycemic control [5,6,7]. Therefore, strategies focused on preserving or restoring β-cell function are vital for effective T2DM treatment.

Pancreatic β-cells have high oxygen consumption during mitochondrial respiration due to the significant energy demands of insulin secretion [2], leading to decreased oxygen levels and hypoxic conditions within the pancreatic tissue. Increasing evidence suggests that hypoxia independently contributes to β-cell dysfunction and accelerates the progression of T2DM [2,8,9]. Hypoxia-inducible factor-1α (HIF-1α) is a critical effector driving the hypoxic phenotype in the body, with its expression upregulated under hypoxic conditions. HIF-1α regulates various physiological processes, including hypoxic adaptation, angiogenesis, immune responses, and energy metabolism [10]. Abnormal lipid metabolism is a common metabolic feature in T2DM patients, primarily characterized by increased levels of free fatty acids (FFA) in the bloodstream [11]. Chronic stimulation by high FFA levels leads to sustained metabolic overload in pancreatic β-cells, inducing lipotoxic effects and further disrupting the cellular redox balance, which is a significant contributor to β-cell hypoxia and subsequent β-cell dysfunction [12,13]. Preliminary evidence suggests that HIF-1α levels are markedly elevated in FFA-treated L02 cells and primary rat hepatocytes compared to controls [14,15]. Furthermore, the knockout of HIF-1α in mouse adipocytes and hepatocytes can improve insulin resistance and glucose intolerance associated with high-fat diets and obesity [16,17]. These studies indicate that HIF-1α, a critical effector molecule of cellular hypoxia, is likely to play a significant role in FFA-induced β-cell dysfunction during T2DM progression. However, the underlying molecular mechanisms remain unclear.

Nicotinamide adenine dinucleotide (NAD^+^), also known as oxidized coenzyme I, is a crucial coenzyme in the tricarboxylic acid cycle, participating in energy synthesis and playing multiple key roles in cellular metabolism, such as energy metabolism, circadian rhythms, aging, longevity control, and so forth [18,19]. Silent information regulator 1 (SIRT1) is a NAD^+^-dependent class III histone deacetylase whose activity and expression are absolutely reliant on NAD^+^ [20]. SIRT1 has been proven to be essential in regulating insulin signaling pathways, lipid metabolism, and glucose homeostasis [21,22,23]. In the pancreatic β-cells of mouse models with diabetes, HIF-1α protein expression is significantly increased, accompanied by a decrease in SIRT1 expression [24,25]. Furthermore, in diabetic nephropathy models, silencing SIRT1 protein promotes fibrosis and inflammation factor expression by upregulating HIF-1α protein, accelerating diabetic nephropathy progression, whereas overexpressing SIRT1 has the opposite effect [26]. These findings indicate that SIRT1, functioning as an upstream regulator of HIF-1α, is probably implicated in β-cell functional impairment caused by FFA-induced hypoxia, but further exploration of the specific mechanism is needed. As a well-known upstream molecule of SIRT1, NAD^+^ expression decreases in cells with Nicotinamide Mononucleotide Adenylyltransferase 1 (NMNAT1) knockout or lactate treatment, inducing pseudohypoxia and increasing HIF-1α protein expression [27]. Additionally, under hypoxic conditions, the stability of HIF-1α in the presence of SIRT1 is influenced by intracellular NAD^+^ levels [28]. This indicates that NAD^+^ regulation of the SIRT1/HIF-1α is crucial in β-cell dysfunction, but the relationship under FFA induction remains to be further explored.

Adenosine monophosphate-activated protein kinase (AMPK) is a crucial biological macromolecule involved in maintaining glucose, lipid, and energy metabolism homeostasis [29]. AMPK regulates cellular energy balance by enhancing ATP-producing catabolic pathways and suppressing energy-consuming processes [30]. AMPK plays a pivotal role in NAD^+^ biosynthesis [31], acting independently of nicotinamide phosphoribosyltransferase (NAMPT), the initial rate-limiting enzyme in the NAD^+^ salvage pathway, to modulate the NAD^+^/NADH ratio and SIRT1 activity [32]. Activating the AMPK pathway can effectively ameliorate high-glucose-induced β-cell dysfunction [33]. In the palmitate (PA, a key component of free fatty acids)-induced diabetic mouse β-cell model, phosphorylated AMPK (p-AMPK) activation increases SIRT1 expression and inhibits β-cell apoptosis [34]. Research indicates that decreased intracellular NAD^+^ levels lead to a pseudohypoxic state, disrupting cellular energy and metabolic balance [27] and resulting in a decreased ADP/ATP ratio that inhibits AMPK expression. Therefore, we hypothesize that elevated NAD^+^ levels may favor AMPK expression within cells. However, whether the NAD^+^/AMPK/SIRT1/HIF-1α pathway is engaged in β-cell dysfunction mediated by FFA-induced hypoxia requires further investigation.

Nicotinamide mononucleotide (NMN), which directly precedes NAD^+^, is a naturally occurring bioactive nucleotide widely found in daily foods such as beans, cabbage, and cucumbers [35]. Recent clinical trials have demonstrated that NMN supplementation is both effective and safe for use in humans [36]. Studies indicate that direct NAD^+^ supplementation can cause side effects such as fatigue, insomnia, and anxiety, whereas NMN supplementation can safely and effectively increase NAD^+^ levels without toxicity [37]. NMN is converted to NAD^+^ via a salvage pathway catalyzed by NMNAT, accounting for up to 85% of the total NAD^+^ synthesis in the body [38]. NMN can effectively improve glucose tolerance in mouse models of T2DM induced by a high-fat diet by restoring NAD^+^ levels and enhancing hepatic insulin sensitivity by activating SIRT1 [39].

Currently, research on NMN’s regulation of β-cell dysfunction is limited, but existing studies suggest that NMN can improve β-cell dysfunction by activating SIRT1 and triggering anti-inflammatory processes [40]. Moreover, NMN intervention can effectively alleviate β-cell dysfunction in conditions of elevated plasma FFA levels [41]. Based on the analysis presented, we propose the following scientific hypothesis: FFA metabolism reduces NAD^+^ levels, inhibiting AMPK and SIRT1 expression, leading to hypoxia-induced HIF-1α protein activation and β-cell dysfunction, thereby disrupting glucose homeostasis and β-cell pathophysiology, ultimately contributing to the development of T2DM; NMN intervention may delay β-cell dysfunction induced by FFA via modulating NAD^+^/AMPK/SIRT1/HIF-1α pathway, thereby improving T2DM.

## 2. Results

### 2.1. PA Impaired Pancreatic β-Cell Function, Reducing Insulin Secretion and NAD^+^/NADH Ratio

We conducted in vitro experiments with INS-1 cells to determine the optimal concentration for establishing a PA-induced pancreatic β-cell dysfunction model. Figure 1A illustrates a dose-dependent reduction in cell viability following PA treatment compared to the control group. Based on prior research and the observed cell viability data, we selected 0.5 mmol/L (mM) PA for 24 h for subsequent experiments [42]. To further validate the feasibility of PA-induced pancreatic β-cell dysfunction, we evaluated insulin secretion levels following a 24 h treatment with 0.5 mM PA in INS-1 cells (Figure 1B), finding that insulin secretion was significantly impaired by 36.09%. Given the crucial role of NAD^+^ during insulin secretion, we observed a significant reduction in the NAD^+^/NADH ratio (29.09%) following the same treatment duration (Figure 1C). Additionally, the protein expression of Glut2, which served as a “glucose sensor”, significantly decreased. However, the metabolic overload induced by PA led to a hypoxic state, which significantly increased the expression of HIF-1α (Figure 1E,F). These results indicate that a 24 h treatment of INS-1 cells with 0.5 mM PA effectively induces dysfunction in pancreatic β-cells, reducing insulin secretion and NAD^+^/NADH ratio.

### 2.2. NMN Alleviated PA-Induced Pancreatic β-Cell Dysfunction, Restoring Insulin Secretion and NAD^+^/NADH Ratio

To investigate the effect of NMN on PA-induced pancreatic β-cell dysfunction, INS-1 cells were subjected to 24 h treatments with either 0.5 mM PA alone or in combination with 0.5 mM NMN. By evaluating insulin secretion and NAD^+^/NADH ratio, we found that the PA+NMN group was able to restore the insulin secretion and NAD^+^/NADH ratio, which were diminished by PA alone (Figure 2A,B). Western blot analysis indicated that NMN treatment inhibited the upregulation of HIF-1α expression and restored Glut2 expression induced by PA (Figure 2D,G), indicating NMN’s efficacy in counteracting PA-induced pancreatic β-cell dysfunction. These findings indicate that NMN holds potential as a promising therapeutic agent for addressing β-cell dysfunction caused by PA.

To gain deeper insight into the molecular mechanisms driving PA-induced pancreatic β-cell dysfunction and assess the effect of NMN, we observed that PA treatment significantly decreased pAMPK and SIRT1 levels while increasing HIF-1α expression. In contrast, NMN treatment restored pAMPK, SIRT1, and Glut2 levels (Figure 2D–F) and inhibited HIF-1α expression (Figure 2G). These findings confirm that NMN’s alleviation of PA-induced pancreatic β-cell dysfunction, with the critical role of pAMPK, SIRT1, and HIF-1α expression.

### 2.3. PA-Induced Pancreatic β-Cell Dysfunction Is Driven by Upregulated HIF-1α Expression

To assess whether HIF-1α expression mediates PA-induced pancreatic β-cell dysfunction, INS-1 cells were exposed to PA alone or in combination with PX-478, an HIF-1α inhibitor, for 24 h. Based on CCK-8 assay results, we selected 20 μM PX-478 for subsequent experiments (Figure 3A). The measurements of insulin secretion and NAD^+^/NADH ratio showed that the PA + PX-478 group displayed 45.16% recovery of insulin secretion and 63.42% recovery of NAD^+^/NADH ratio compared to the PA group (Figure 3B,C). Western blot results indicated that, compared to the PA group, PX-478 inhibited HIF-1α expression and increased Glut2 expression, while pAMPK and SIRT1 expression remained unaffected (Figure 3E–H). These findings indicate that the dysfunction of pancreatic β-cells induced by PA is driven by elevated expression of HIF-1α.

### 2.4. PA-Induced Pancreatic β-Cell Dysfunction Is Related to NAD^+^/AMPK/SIRT1/HIF-1α Signaling Pathway

The results above provide preliminary evidence linking PA-induced pancreatic β-cell dysfunction to increased HIF-1α expression. However, the interactions between AMPK, SIRT1, and HIF-1α have yet to be thoroughly investigated. To explore the essential role of AMPK in PA-induced pancreatic β-cell dysfunction, INS-1 cells were exposed to PA and/or AICAR (an AMPK activator) for 24 h. Considering prior studies and the outcomes of the CCK8 cell viability assay, we selected 0.5 mM as the intervention concentration for AICAR (Figure 4A) [43,44]. By detecting insulin secretion and NAD^+^/NADH ratio after intervention, we observed that, in contrast to the PA group, the PA + AICAR treatment led to a recovery of insulin secretion by 43.04% and an improvement in the NAD^+^/NADH ratio by 260.93% (Figure 4B,C). Additionally, Western blot results indicated that the PA + AICAR treatment resulted in elevated levels of pAMPK, SIRT1, and Glut2 expression, accompanied by decreased HIF-1α expression (Figure 4E–H). These results suggest that the dysfunction of pancreatic β-cells induced by PA is primarily driven by the activation of pAMPK. The NAD^+^/AMPK/SIRT1/HIF-1α signaling pathway is activated by PA, leading to the occurrence of pancreatic β-cell dysfunction.

### 2.5. NMN Mitigates PA-Induced Pancreatic β-Cell Dysfunction by Reducing HIF-1α Expression

To explore how NMN influences pancreatic β-cell dysfunction by targeting HIF-1α, INS-1 cells were exposed to 0.5 mM PA, either alone or in conjunction with 0.5 mM NMN and/or DFO (an HIF-1α activator), for 24 h. Based on CCK8 results, we selected 0.2 mM as the intervention concentration of DFO (Figure 5A). In the GSIS assay, the results revealed that treatment with PA + NMN + DFO led to a significant reduction in insulin secretion capacity in INS-1 cells, showing a decrease of 36.18% compared to the PA + NMN treatment group (Figure 5B). Moreover, compared to the PA + NMN group, the PA + NMN + DFO treatment displayed a marked decrease (42.50%) in the NAD^+^/NADH ratio (Figure 5C). Western blot results showed that NMN treatment restored PA-induced pAMPK, and SIRT1, Glut2 expression and inhibited HIF-1α expression, whereas the beneficial effects of NMN were reversed by the HIF-1α activator DFO (Figure 5E–H). These findings indicate that the alleviation of PA-induced pancreatic β-cell dysfunction by NMN treatment is mediated through HIF-1α.

### 2.6. NMN Alleviates PA-Induced Pancreatic β-Cell Dysfunction by Modulating the NAD^+^/AMPK/SIRT1/HIF-1α Signaling Pathway

Our results confirm the activation of the NAD^+^/AMPK/SIRT1 pathway is associated with PA-induced pancreatic β-cell dysfunction mediated by HIF-1α. Nonetheless, it remains to be determined whether the inhibitory effect of NMN on pancreatic β-cell dysfunction operates through the NAD^+^/AMPK/SIRT1 pathway. INS-1 cells were exposed to 0.5 mM PA alone or in combination with 0.5 mM NMN and/or CC (an AMPK inhibitor) for 24 h. CCK-8 assay results showed that CC had no cytotoxicity on INS-1 cells at concentrations below 10 μM (Figure 6A). Based on the literature and CCK-8 results, we selected 10 μM CC for subsequent experiments [45]. In the GSIS assay, the PA + NMN + CC group exhibited significantly reduced insulin secretion levels (54.18%) compared to the PA + NMN group (Figure 6B). Additionally, the NAD^+^/NADH ratio in INS-1 cells was significantly decreased by 50.60% in the PA + NMN + CC group compared to the PA + NMN group (Figure 6C). Western blot results showed that NMN restored PA-induced reductions in pAMPK and SIRT1 expression and inhibited HIF-1α expression. However, in the PA + NMN + CC group, HIF-1α expression increased while pAMPK, SIRT1, and Glut2 expression decreased compared to the PA + NMN group (Figure 6E–H). These results demonstrate that CC significantly counteracts the beneficial effects of NMN on PA-induced pancreatic β-cell dysfunction. Additionally, our findings further validate that the NMN-mediated improvement in PA-induced pancreatic β-cell dysfunction is largely contingent upon its modulation of the NAD^+^/AMPK/SIRT1/HIF-1α signaling pathway.

## 3. Discussion

This study aims to elucidate the mechanisms underlying β-cell dysfunction induced by hypoxia in response to PA and to investigate the potential mitigating effects of NMN. To our knowledge, our research is the first to demonstrate that PA induces β-cell dysfunction via inhibiting the NAD^+^/AMPK/SIRT1/HIF-1α pathway, whereas NMN treatment can reverse PA-induced β-cell dysfunction by activating this pathway, providing an emerging perspective for the therapy of T2DM.

HIF-1α plays a pivotal role in regulating oxygen homeostasis as part of the body’s adaptive response to hypoxic conditions [46]. However, the role of HIF-1α in PA-induced pancreatic β-cell dysfunction is not well elucidated. In our study, we found that PA treatment led to an increase in HIF-1α expression, which was associated with a reduction in insulin secretion function. Notably, PX-478 supplementation effectively reversed PA-induced β-cell dysfunction. In line with our observations, prior research has reported a high presence of HIF-1α-positive β-cells in diabetic mouse models [8], and PX-478 treatment has demonstrated efficacy in enhancing β-cell function in individuals with diabetes [2]. Our findings, in line with multiple studies, suggest that inhibiting HIF-1α expression may be a crucial strategy for restoring β-cell function [2,47,48], suggesting that PA-induced β-cell dysfunction is closely associated with the upregulation of HIF-1α.

Concurrently, we identified that the disruption of energy homeostasis in PA-induced pancreatic β-cell dysfunction led to a decreased NAD^+^/NADH ratio and inhibited the activation of AMPK/SIRT1 [49]. However, supplementation with the AMPK activator AICAR effectively reversed the PA-induced β-cell dysfunction. Previous research has demonstrated that pharmacological activation of AMPK with AICAR markedly enhances insulin secretion in rat islets [50]. Additionally, studies involving perfused rat pancreas have corroborated that AICAR-induced AMPK activation substantially boosts insulin secretion [50,51]. SIRT1, a critical target of AMPK in various tissues [52], responds to changes in intracellular energy status, as reflected by the NAD^+^/NADH ratio. The absence of SIRT1 has been demonstrated to lead to compromised insulin secretion in islets [49,53]. Moreover, treatment with resveratrol has been shown to increase insulin secretion in human islets in a SIRT1-dependent manner, accompanied by increased pAMPK and Glut2 protein expression [54]. Our findings indicate that AMPK is probably a critical regulator in the development of pancreatic β-cell dysfunction induced by PA.

NMN, a direct precursor in NAD^+^ biosynthesis, has recently demonstrated significant promise in preventing and treating aging, metabolic disorders, and neurodegenerative diseases [39,55,56,57]. Research has shown that NMN supplementation enhances pancreatic β-cell function and boosts insulin sensitivity by increasing NAD^+^ levels, thereby preventing or ameliorating T2DM [39]. Furthermore, NMN aids in the prevention of obesity and metabolic syndrome by modulating metabolic pathways, enhancing lipolysis, and improving insulin sensitivity [55]. Despite the promising potential of NMN, its effects on PA-induced β-cell dysfunction have not been extensively studied. To explore this, we treated INS-1 cells subjected to PA with NMN. Our results demonstrated that NMN supplementation significantly counteracted PA-induced β-cell dysfunction, restored the NAD^+^/NADH ratio, and improved insulin secretion. Additionally, NMN treatment normalized the expression levels of key proteins involved. These findings underscore NMN’s effectiveness and its potential as a therapeutic agent for β-cell dysfunction and T2DM.

The precise role of NMN in PA-induced islet β-cell dysfunction is not yet fully understood. To elucidate how NMN influences this dysfunction, we investigated its impact on the expression of HIF-1α and AMPK in the context of PA treatment. Our study demonstrated that the intervention with the AMPK inhibitor CC enhances the alleviating effect of NMN on islet β-cell dysfunction, while supplementation with the HIF-1α activator DFO negates this beneficial effect. In hepatocellular carcinoma (HCC), NMN supplementation notably enhanced the p-AMPK/AMPK ratio, which in turn promotes autophagy and ferroptosis via activating the AMPK/mTOR signaling pathway, thereby contributing to the suppression of HCC progression [58]. Furthermore, in brown adipose tissue of aging-accelerated mice, NMN supplementation enhanced both oxidative stress responses and SIRT1 protein levels in the brown adipose tissue of mice with accelerated aging [59]. The activity of SIRT1, a deacetylase enzyme, relies on NAD^+^ availability, which is quickly synthesized from NMN once it is absorbed by cells [60]. As intracellular NAD^+^ levels and energy status increase, the activities of AMPK and SIRT1 are also enhanced. Activated SIRT1 directly enhances the expression and function of AMPK by deacetylating transcription factors such as PGC-1α and indirectly through deacetylation of LKB1, an upstream kinase of AMPK [61]. In hypoxic tissues of mice, NMN supplementation has been shown to attenuate HIF-1α activation and fibrosis by activating the NAD^+^/SIRT1 axis [62].

To our knowledge, this is the first to investigate the therapeutic strategy of NMN in regulating PA-induced pancreatic β-cell dysfunction via the NAD^+^/AMPK/SIRT1/HIF-1α pathway. However, this study only examined the effects of AMPK and HIF-1α activators and inhibitors. Future research should explore the roles of various proteins in this signaling pathway through techniques such as gene silencing and overexpression. Additionally, our study lacks in vivo validation, highlighting the need for further investigation.

## 4. Materials and Methods

### 4.1. Cell Culture and Treatments

INS-1 rat insulinoma cells were acquired from ATCC and cultured in RPMI 1640 medium (Titan Scientific, Shanghai, China) supplemented with 10% fetal bovine serum (FBS) (Gibco, Grand Island, NE, USA), 50 μmol/L β-mercaptoethanol (Sigma-Aldrich, St. Louis, MO, USA), 1 mM sodium pyruvate, 10 mM HEPES, 100 U/mL penicillin and 100 μg/mL streptomycin (Titan Scientific, Shanghai, China) at 37 °C in a 5% CO_2_ incubator. NMN was obtained from APExBIO (Boston, MA, USA). PA was purchased from MACHLIN (Shanghai, China). PX-478 (a HIF-1α inhibitor, Cat No. HY-10231) and Deferoxamine mesylate (DFO, an HIF-1α activator, Cat No. HY-B0988) were obtained from MedChemExpress (Monmouth Junction, NJ, USA). AICAR (an AMPK activator, Cat No. B1211) and Compound C (CC, an AMPK, Cat No. B3252) were obtained from APExBIO.

### 4.2. Cell Viability Assay

INS-1 cell viability was assessed using the Cell Counting Kit-8 (CCK-8, APExBIO) after treatment with different concentrations of PA (0, 0.25, 0.5, 0.75, 1.00 mM), PX-478 (0, 10, 20, 30, 40 μmol/L), DFO (0, 0.1, 0.2, 0.3, 0.4 mM), AICAR (0, 5, 10, 15, 20 μmol/L), and CC (0, 0.25, 0.5, 0.75, 1.00 mM) for 24 h.

### 4.3. Glucose-Stimulated Insulin Secretion Assay (GSIS) and Measurement of NAD^+^/NADH Ratio

INS-1 cells were seeded into 24-well plates and treated with optimal concentrations of PA alone or with NMN, PX-478, AICAR, DFO, and CC for 24 h on the following day. Then, the medium was replaced with freshly glucose-free Krebs–Ringer bicarbonate HEPES buffer (KRB buffer containing 135 mM NaCl, 3.6 mM KCl, 0.5 mM NaH_2_PO_4_, 2 mM NaHCO_3_, 1.5 mM CaCl_2_, 0.5 mM MgSO_4_, 10 mM HEPES and 0.1% BSA (bovine serum albumin)) for 1 h. After this, INS-1 cells were incubated with fresh KRB buffer containing 3.3 mM or 16.7 mM glucose for 1 h. The 3.3 mM glucose concentration simulates normal fasting glucose levels, while the 16.7 mM glucose concentration reflects postprandial hyperglycemia. This contrast enables us to effectively assess the insulin secretion responses of pancreatic β-cells under varying glucose conditions, thereby elucidating functional changes in these cells across different physiological states. Insulin levels in the cell culture supernatant were quantified using an ELISA kit (Elabscience Biotechnology, Wuhan, China) following the manufacturer’s protocol. Insulin secretion levels were normalized to the total cellular protein content in INS-1 cells. Following 24 h treatments with PA alone or in combination with NMN, PX-478, DFO, AICAR, and CC, the NAD^+^/NADH ratio in INS-1 cells was quantified using the Coenzyme I NAD(H) Content Assay Kit (Boxbio, Beijing, China) according to the manufacturer’s instructions.

### 4.4. Western Blot Analysis

INS-1 cells subjected to various treatments were harvested and lysed using RIPA lysis buffer. Equal protein amounts from the lysates were resolved on 10% SDS-PAGE gels and transferred onto polyvinylidene difluoride (PVDF) membranes (Millipore, MA, USA) via electrophoresis. The membranes were then blocked with 5% non-fat dry milk in Tween-Tris-buffered saline and incubated overnight at 4 °C with the respective primary antibodies: GAPDH (ABclonal Technology, Wuhan, China), anti-Glut2 (ABclonal), anti-AMPKa1/AMPKa2 (ABclonal), anti-phospho-AMPKa1/AMPKa2-T183/T172 (ABclonal), anti-SIRT1 (CST, Danvers, MA, USA) and anti-HIF-1α (Boster Biological Technology, Wuhan, China). Then, the membranes were incubated with horseradish peroxidase-linked secondary antibody: horseradish peroxidase (HRP)-linked anti-rabbit IgG (Beyotime Biotechnology, Shanghai, China) and HRP-linked anti-mouse IgG (Beyotime). Immunoreactive bands were visualized using ECL detection reagent (ShareBio, Shanghai, China), and their optical densities were quantified with Image J 1.8.0 software.

### 4.5. Statistical Analysis

Data from repeated experiments are presented as mean ± SD. Statistical analysis was performed using SPSS 25.0 software, and significance was defined as *p* < 0.05. The Student’s *t*-test was used to compare the difference between two groups. Multiple comparisons involving three or more groups were evaluated by one-way ANOVA followed by post hoc Tukey test.

## 5. Conclusions

Our study presents novel in vitro evidence underscoring the pivotal role of FFA in inducing pancreatic β-cell dysfunction and underscores the protective benefits of NMN. Specifically, NMN was shown to counteract the FFA-induced activation of the NAD^+^/AMPK/SIRT1/HIF-1α pathway, thereby alleviating β-cell dysfunction. By elucidating the involvement of this signaling pathway in β-cell dysfunction, our research provides novel perspectives into NMN’s potential for preventing T2DM and suggests possible therapeutic targets for future interventions.

## Figures and Tables

**Figure 1 ijms-25-10534-f001:**
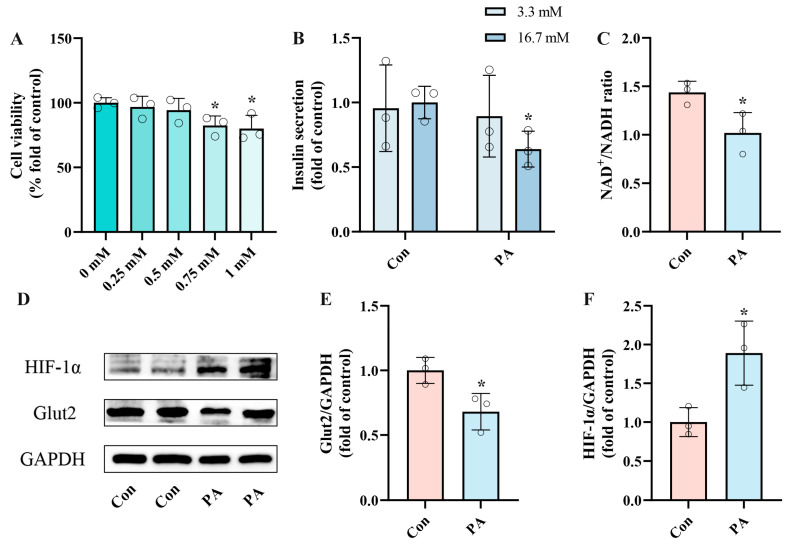
PA induced pancreatic β-Cell dysfunction, impairing insulin secretion and reducing NAD^+^/NADH ratio. (**A**) Cell viability of INS-1 cells treated by different PA concentrations for 24 h. (**B**) The secretion level of INS-1 cells treated with 0.5 mM PA. Measurement 3.3 mM represents 3.3 mM glucose concentration simulating normal fasting glucose levels, while 16.7 mM represents 16.7 mM glucose concentration reflecting postprandial hyperglycemia. Insulin levels were measured after stimulation with 3.3 mM glucose and 16.7 mM glucose after 1 h of incubation. (**C**) The NAD^+^/NADH ratio of INS-1 cells treated by 0.5 mM PA. (**D**) Representative Western blotting of HIF-1α, Glut2, and GAPDH in INS-1 cells. (**E**,**F**) Quantitative density analysis of Glut2 and HIF-1α normalized to GAPDH. Data are expressed as mean ± SD (*n* = 3). Each circle represents a replicate experiment. * *p* < 0.05 vs. the control group.

**Figure 2 ijms-25-10534-f002:**
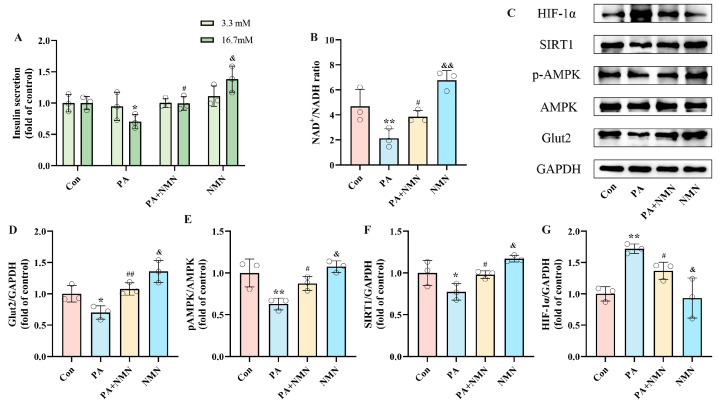
NMN alleviated PA-induced pancreatic β-cell dysfunction, restoring insulin secretion and increasing NAD^+^/NADH ratio. (**A**) The secretion level of INS-1 cells treated with 0.5 mM PA and/or 0.5 mM NMN. Measurement 3.3 mM represents 3.3 mM glucose concentration simulating normal fasting glucose levels, while 16.7 mM represents 16.7 mM glucose concentration reflecting postprandial hyperglycemia. Insulin levels were measured after stimulation with 3.3 mM glucose and 16.7 mM glucose after 1 h of incubation. (**B**) The NAD^+^/NADH ratio of INS-1 cells treated by 0.5 mM PA and/or 0.5 mM NMN. (**C**) Representative Western blotting of HIF-1α, SIRT1, pAMPK, AMPK, Glut2, and GAPDH in INS-1 cells. (**E**) p-AMPK/AMPK ratio. (**D**,**F**,**G**) Quantitative density analysis of Glut2, SIRT1, and HIF-1α normalized to GAPDH. Data are expressed as mean ± SD (*n* = 3). Each circle represents a replicate experiment. * *p* < 0.05, ** *p* < 0.01 vs. the control group; ^#^ *p* < 0.05, ^##^ *p* < 0.01 vs. the PA group; ^&^ *p* < 0.05, ^&&^ *p* < 0.01 vs. the PA + NMN group.

**Figure 3 ijms-25-10534-f003:**
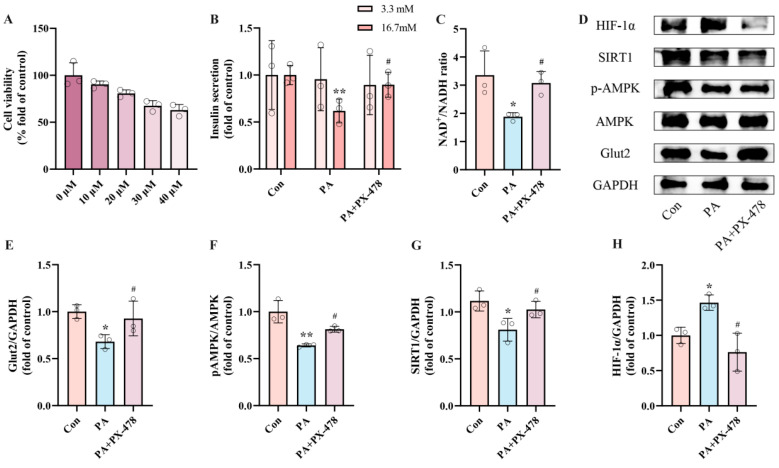
PA-induced pancreatic β-cell dysfunction is mediated by upregulated HIF-1α expression. (**A**) Cell viability of INS-1 cells treated by different PX-478 concentrations. (**B**) The secretion level of INS-1 cells treated by 0.5 mM PA and/or 20 μM PX-478. Measurement 3.3 mM represents 3.3 mM glucose concentration simulating normal fasting glucose levels, while 16.7 mM represents 16.7 mM glucose concentration reflecting postprandial hyperglycemia. Insulin levels were measured after stimulation with 3.3 mM glucose and 16.7 mM glucose after 1 h of incubation. (**C**) The NAD^+^/NADH ratio of INS-1 cells treated by 0.5 mM PA and/or 20 μM PX-478. (**D**) Representative Western blotting of HIF-1α, SIRT1, pAMPK, AMPK, Glut2, and GAPDH in INS-1 cells. (**F**) p-AMPK/AMPK ratio. (**E**,**G**,**H**) Quantitative density analysis of Glut2, SIRT1, and HIF-1α normalized to GAPDH. Data are expressed as mean ± SD (*n* = 3). Each circle represents a replicate experiment. * *p* < 0.05, ** *p* < 0.01 vs. the control group; ^#^ *p* < 0.05 vs. the PA group.

**Figure 4 ijms-25-10534-f004:**
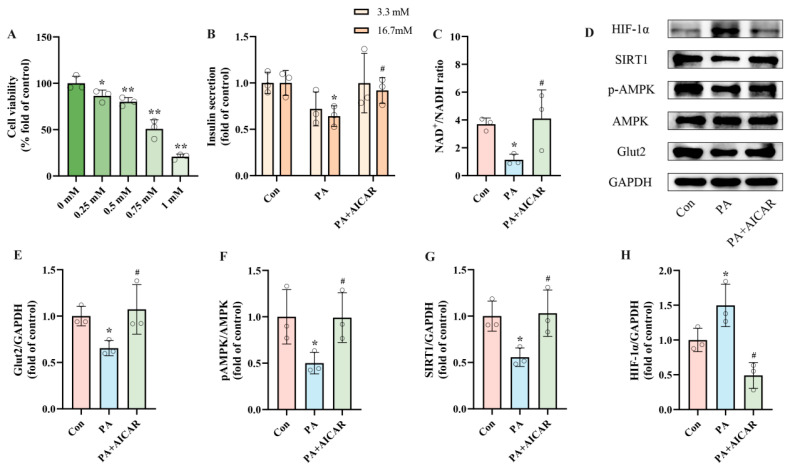
PA-induced pancreatic β-cell dysfunction is related to NAD^+^/AMPK/SIRT1/HIF-1α signaling pathway regulation. (**A**) Cell viability of INS-1 cells treated by different AICAR concentrations. (**B**) The secretion level of INS-1 cells treated with 0.5 mM PA and/or 0.5 mM AICAR. Measurement 3.3 mM represents 3.3 mM glucose concentration simulating normal fasting glucose levels, while 16.7 mM represents 16.7 mM glucose concentration reflecting postprandial hyperglycemia. Insulin levels were measured after stimulation with 3.3 mM glucose and 16.7 mM glucose after 1 h of incubation. (**C**) The NAD^+^/NADH ratio of INS-1 cells treated by 0.5 mM PA and/or 0.5 mM AICAR. (**D**) Representative Western blotting of HIF-1α, SIRT1, pAMPK, AMPK, Glut2, and GAPDH in INS-1 cells. (**F**) p-AMPK/AMPK ratio. (**E**,**G**,**H**) Quantitative density analysis of Glut2, SIRT1, and HIF-1α normalized to GAPDH. Data are expressed as mean ± SD (*n* = 3). Each circle represents a replicate experiment. * *p* < 0.05, ** *p* < 0.01 vs. the control group; ^#^ *p* < 0.05 vs. the PA group.

**Figure 5 ijms-25-10534-f005:**
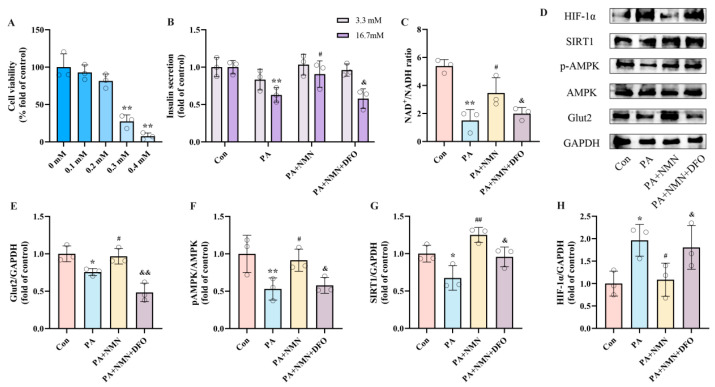
NMN alleviates PA-induced pancreatic β-cell dysfunction by inhibiting HIF-1α expression. (**A**) Cell viability of INS-1 cells treated by different DFO concentrations. (**B**) The secretion level of INS-1 cells treated with 0.5 mM PA and/or 0.5 mM NMN and/or 0.2 mM DFO. Measurement 3.3 mM represents 3.3 mM glucose concentration simulating normal fasting glucose levels, while 16.7 mM represents 16.7 mM glucose concentration reflecting postprandial hyperglycemia. Insulin levels were measured after stimulation with 3.3 mM glucose and 16.7 mM glucose after 1 h of incubation. (**C**) The NAD^+^/NADH ratio of INS-1 cells treated by 0.5 mM PA and/or 0.5 mM NMN and/or 0.2 mM DFO. (**D**) Representative Western blotting of HIF-1α, SIRT1, pAMPK, AMPK, Glut2, and GAPDH in INS-1 cells. (**F**) p-AMPK/AMPK ratio. (**E**,**G**,**H**) Quantitative density analysis of Glut2, SIRT1, and HIF-1α normalized to GAPDH. Data are expressed as mean ± SD (*n* = 3). Each circle represents a replicate experiment. * *p* < 0.05, ** *p* < 0.01 vs. the control group; ^#^ *p* < 0.05, ^##^ *p* < 0.01 vs. the PA group; ^&^ *p* < 0.05, ^&&^ *p* < 0.01 vs. the PA + NMN group.

**Figure 6 ijms-25-10534-f006:**
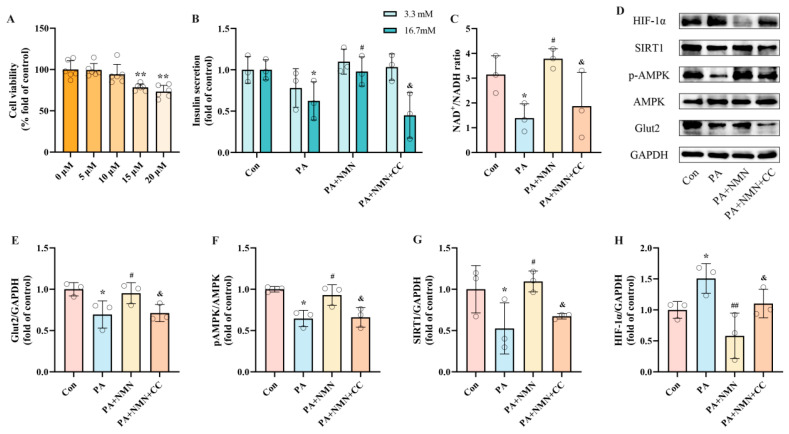
NMN alleviates PA-induced pancreatic β-cell dysfunction by modulating the NAD^+^/AMPK/SIRT1/HIF-1α signaling pathway. (**A**) Cell viability of INS-1 cells treated by different CC concentrations. (**B**) The secretion level of INS-1 cells treated by 0.5 mM PA and/or 0.5 mM NMN and/or 10 μM CC. Measurement 3.3 mM represents 3.3 mM glucose concentration simulating normal fasting glucose levels, while 16.7 mM represents 16.7 mM glucose concentration reflecting postprandial hyperglycemia. Insulin levels were measured after stimulation with 3.3 mM glucose and 16.7 mM glucose after 1 h of incubation. (**C**) The NAD^+^/NADH ratio of INS-1 cells treated by 0.5 mM PA and/or 0.5 mM NMN and/or 10 μM DFO. (**D**) Representative Western blotting of HIF-1α, SIRT1, pAMPK, AMPK, Glut2, and GAPDH in INS-1 cells. (**F**) p-AMPK/AMPK ratio. (**E**,**G**,**H**) Quantitative density analysis of Glut2, SIRT1, and HIF-1α normalized to GAPDH. Data are expressed as mean ± SD (*n* = 3). Each circle represents a replicate experiment. * *p* < 0.05, ** *p* < 0.01 vs. the control group; ^#^ *p* < 0.05, ^##^ *p* < 0.01 vs. the PA group; ^&^ *p* < 0.05 vs. the PA + NMN group.

## Data Availability

Data are contained within the article.

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
