# Peer review of "Nicotinamide Mononucleotide (NMN) Ameliorates Free Fatty Acid-Induced Pancreatic β-Cell Dysfunction via the NAD+/AMPK/SIRT1/HIF-1α Pathway"

_ijms, 2024, doi:10.3390/ijms251910534_

Round 1
Reviewer 1 Report
Comments and Suggestions for Authors
Comments on Nicotinamide mononucleotide (NMN) ameliorates free fatty acid-induced pancreatic β-cell dysfunction via the NAD+/AMPK/SIRT1/HIF-1α pathway
In the manuscript, the authors present an in-depth investigation into the mechanism of free fatty acid-induced pancreatic beta cell dysfunction through NAD+/AMPK/SIRT1/HIF-1α pathway in INS-1 cells. They further explore the mechanism of NMN drug in ameliorating beta cell dysfunction by counteracting the same pathway. Overall, the methods employed are comprehensive with the use of difference activators and inhibitors along the pathway, and the results are convincing.
I would recommend acceptance of the manuscript, provided the following suggestions and questions are addressed.
1. In the abstract, please include the full term of palmitate (PA) before using the abbreviation.
2. It would be beneficial if a schematic of the investigated mechanism pathway was included.
3. In all insulin secretion column chart, could you elaborate in the text or figure legend on what 3.3 mM and 16.7 mM indicate? Why were these specific concentrations chosen?
4. In figure 1d and 2c, please include lane annotations for the western blots.
5. In figure 2c, why are there two bands for Glut2, while in other western blots there is only one band? Would suggest repeat western blot to avoid confusion.
6. Is there a cytotoxicity assay to demonstrate the safety profile of NMN in this study? Additionally, why was 0.5 mM NMN chosen for cell treatment?
Reviewer 2 Report
Comments and Suggestions for Authors
The article “Nicotinamide mononucleotide (NMN) ameliorates free fatty acid-induced pancreatic β-cell dysfunction via the NAD+/AMPK/SIRT1/HIF-1α pathway” by Wang et al. addresses an exciting topic. The authors attempt to demonstrate that NMN reverses AP-induced dysregulation of the NAD+/AMPK/SIRT1/HIF-1α pathway, ameliorating β-cell dysfunction. The article is well written, and the methodology seems appropriate, but several concerns need further clarification.
Majors
1. My main concern is the statistical analysis. The authors mention that they performed a one-way ANOVA followed by the Student-Newman-Keuls post hoc test. However, how did they perform ANOVA on the results in Figures 1C, E, and F? There are only two groups. It would be better to do a t-student test instead of ANOVA.
2. The Newman-Keuls test works well with three groups. However, the Type I error increases when four or more groups are compared. Considering this, this test could also be misapplied in Figure 1A. The same occurs in Figures 2, 5, and 6, where there are more than three groups.
3. Another critical point is that in all the figures, a graph appears (Figure 1B, 2A, 3B, 5B, 6B) with two concentrations, 3.3.mM, and 16.7 mM; these data are not described in any part of the article, confusing understanding these experiments.
4. The authors should perform a proper statistical analysis.
Minors
1. In Figures 1 and 2, the western blot does not have the names in the lanes.
2. Figure 4 does not exist.
3. At the bottom of Figure 1, they put * P < 0.05, ** P < 0.01 vs. the control group; # P < 0.05. But there are no symbols in the graphs.
Round 2
Reviewer 2 Report
Comments and Suggestions for Authors
The authors responded adequately to all my concerns